# Estimating the Prevalence and Factors Affecting the Shedding of Helminth Eggs in Irish Equine Populations

**DOI:** 10.3390/ani13040581

**Published:** 2023-02-07

**Authors:** Nagwa Elghryani, Trish McOwan, Craig Mincher, Vivienne Duggan, Theo de Waal

**Affiliations:** 1Telenostic Ltd., R95 CRK2 Dublin, Ireland; 2Department of Biology, Faculty of Arts and Sciences-Gamines, University of Benghazi, Benghazi 33FX+QV9, Libya; 3School of Veterinary Medicine, University College Dublin, D04 D6F6 Dublin, Ireland

**Keywords:** gastrointestinal helminths, faecal egg count (FEC), prevalence, risk factors, Ireland

## Abstract

**Simple Summary:**

Gastrointestinal helminths are the most important parasite group infecting horses and donkeys throughout the world. There is limited information on the prevalence of infection with these parasites in Ireland. This study attempted to investigate the prevalence and diversity of helminth parasites in Irish equine populations and identify factors associated with helminth egg shedding. We observed that the prevalence of infection with strongyle parasites was 52.4% in horses on Irish farms, and the prevalence of infection ranged from 5% to 100% in horses kept under differing management practices. In addition, it was established that young horses had the highest faecal output of helminth eggs, which needs to be considered in future sustainable control programs.

**Abstract:**

Gastrointestinal helminths are ubiquitous in horse populations across the world. Intensive anthelmintic treatments have succeeded in controlling some of the pathogenic effects of these parasites. However, the success of anthelmintic drugs has been threatened by the development of widespread resistance to those most commonly used. To develop improved control strategies, information on helminth distribution patterns is needed, which can be obtained by identifying animals regarded as high egg shedders and taking age, gender, and other risk factors into account. The aim of this study was to determine the prevalence of helminth infection in the Irish equine population using faecal egg counts and to evaluate the effects of risk factors on these faecal egg counts. For the 2700 horses that were included in the study, the prevalence of gastrointestinal helminth infection was 52.40%, 4.22%, 2.59%, and 0.89% for strongyle species, *Parascaris* spp., *Anoplocephala* spp., and *Strongyloides westeri*, respectively. Overall, strongyle faecal egg counts from 159 farms averaged 250.22 eggs per gram. Both age and season had significant effects on strongyle egg shedding. In conclusion, this study revealed high prevalence of strongyle worm infection in horses on Irish farms, which highlights the need to optimize and develop good management practices and strategic deworming.

## 1. Introduction

Gastrointestinal helminths (GIHs) are a major constraint on the health and welfare of horses and donkeys [1,2,3]. Horses are commonly infected with large and small strongyles, *Parascaris equorum, P. univalens, Strongyloides westeri, Anoplocephala* spp., and *Oxyuris equi* in many countries under different climatic conditions [3,4,5].

Large strongyles—in particular, *Strongylus vulgaris*—have traditionally been regarded as the most important and pathogenic parasites of horses and have a high mortality rate [3,6,7]. However, with the introduction of modern broad-spectrum anthelmintics, such as the macrocyclic lactones (MLs), in the 1980s, large strongyles have been effectively controlled with routine deworming programs. This has resulted in small strongyles (cyathostomins) coming to the fore. Partly due to the continued and frequent use of anthelmintic drugs, anthelmintic resistance (AR) has become a major issue in this parasite population. Large numbers of cyathostomin larvae can potentially encyst in the gut wall, and most anthelmintic drugs are not very effective at treating them [8]. Later, these larvae will re-activate and excyst. If this emergence of encysted larvae occurs en masse, it can cause serious damage to the gut wall, resulting in an inflammatory syndrome affecting the caecum and colon. In severe cases, this may lead to a clinical syndrome called acute larval cyathostominosis, which can be fatal [9,10,11,12,13]. In addition, horses with heavy small strongyle burdens suffer from weight loss and show poor conditions and growth.

In addition to small strongyles, *Parascaris* spp. and *S. westeri* are the most prevalent species in foals and yearlings [14,15,16,17]. Since benzimidazole (BZ) resistance has become widespread among cyathostomins, MLs are being used much more widely, contributing to the development of AR in *Parascaris* spp. to MLs [18].

It is generally accepted that only a small proportion of animals carry a high parasite burden, and they are, therefore, an important source of environmental contamination with helminth eggs [19,20]. Despite their drawbacks, faecal egg counts (FECs) are still a useful monitoring tool that can be used to target animals with high FECs for selective treatment strategies (STSs) [21,22,23,24,25,26,27] to reduce contamination of pasture. Despite the development of strategies for the minimization of the development of anthelmintic resistance, the development of novel FEC tools for use in livestock should remain a priority. Accordingly, current research efforts have been aimed at developing diagnostic tools based on FECs, which are the cornerstone of STSs [28,29,30,31,32].

Moreover, strongyle infections and shedding of helminth eggs have been found to be affected by many factors, such as the type of stud, breed of horse, age, gender, management type, pasture type, and season, as shown in several studies that have been carried out on the distribution and the prevalence of helminths worldwide [3,20,33,34,35,36,37,38,39]. This study was carried out in Ireland with the objectives of determining the prevalence and diversity of helminths in Irish equine populations using FECs and evaluating the effects of age, gender, season, and region on FECs.

## 2. Materials and Methods

### 2.1. Sample and Data Collection

To examine the presence of helminth eggs in Irish equine populations, a total of 2700 horses were randomly selected from 159 horse farms. Faecal samples were collected over two periods of time. Initially, 964 faecal samples were collected from 29 horse farms between January 2015 and December 2017. Then, 1736 faecal samples from 130 horse farms were collected between July 2019 and November 2021. The farms were located in different regions of Ireland (Figure 1): the north and western region (95 horses), the southern region (375 horses), and the eastern and midland regions (928 horses). The owner was asked to collect a fresh faecal sample from each individual horse. Each faecal sample was placed into a labelled container before being sent to the laboratory, where it was stored at 4 °C upon arrival. All samples were processed within three days of collection. Forms were completed for individual samples with information on the age, sex, season when the sample was collected, geographical area of the farm, and history of treatment against parasites.

To study age, 1546 animals were separated into three age groups: up to 5 years, 5–17 years, and >17 years. To study gender, 2057 animals were included in this study: 55.9% were female and 44.1% male (23.1% fillies, 20.1% colts, 36.6% adult females, and 20.2% adult males). To study seasons, 2652 animals were included: 26.8% had samples collected in the winter, 15.1% in spring, 14.2% in summer, and 43.9% in autumn.

### 2.2. Faecal Egg Count Analysis

The FEC analysis was carried out using a quantitative modified Mini-FLOTAC technique with a minimum detection limit of 5 eggs per gram (EPG) [29,40]. All faecal samples were individually processed. Each faecal sample was well-homogenized, with 5 g thoroughly mixed in 45 mL of saturated sodium chloride solution (NaCl; specific gravity: 1.200). The suspension was filtered through a tea-strainer, and the filtered slurry was loaded into each of the chambers of the Mini-FLOTAC (University of Naples Federico II, Naples, Italy). The slide was left for 10 min before being examined under a microscope (Optika Microscope model B-800BF) at 10x magnification. All the eggs were counted in both chambers. The total egg count in the two chambers was multiplied by five to estimate the number of EPG for strongyles, *Parascaris* spp., *Anoplocephala* spp., and *S. westeri* individually.

### 2.3. Data Analysis

Statistical analysis was performed using IBM SPSS (Version 27, 2020). The mean of the FECs was calculated as the average of all the recorded EPG, including zero EPG, while the mean intensity of the EPG for each parasite species was the mean number of the EPG found in the infected horses [41]. To investigate the aggregated distribution [42] of egg excretion by hosts, the number of horses that were responsible for excreting 80% of the eggs was calculated in an Excel (Microsoft Office 2013) datasheet from the total EPG for all horses with a positive FEC and expressed as a percentage.

The effects of age, gender, sampling season, and region of sample collection on helminth egg shedding were determined using faecal samples from 159 horse farms. Prior to analysis, the data were evaluated for normal distribution with Kolmogorov–Smirnov, Shapiro–Wilk, and normal probability plots. As the results showed that none of the datasets were normally distributed, the generalized linear model univariate test was used as it can handle empty cell problems when estimating each marginal mean of the dependent variables (strongyle EPG), and only cases with no missing data were used [21,43]. When the overall *F* test showed the significance of a factor, a pairwise comparison of the estimated marginal means of the dependent variables (strongyle EPG) was used to evaluate the differences between specific means of EPG for different factors. The mean was estimated according to the estimated marginal mean, which was the mean of one variable averaged across every combination of the levels of the other variables in the model, and interaction plots of these means are presented to allow easy visualization of the relationships. The standard error and confidence intervals for the mean were based on 1000 bootstraps. The significance level was set at *p* < 0.05.

## 3. Results

The prevalence of strongyle worm eggs was 52.4% (1415/2700), with a mean of 250.22 EPG (S.E. = 12; 95% CI: 228–275) and a mean intensity of 477.46 EPG (S.E. = 20.43; 95% CI: 437.66–520.89). More than half of the infected animals (51.20%; 725/1415; 95% CI: 48.90–54.00) from 110 farms had strongyle FECs > 200 EPG, while 52 farms had horses with mean strongyle FECs > 500 EPG, which indicated a significant difference (*p* < 0.001).

The prevalence of *Parascaris* spp., *Anoplocephala* spp., and *S. westeri* egg excretion was very low (Table 1).

Regarding the rule of 80/20, 80% of all EPG values recorded (540,467.21) were shed by 32.08% of the horses, in which the EPG ranged from 457.50 to 13,282.50.

Due to missing information for some animals, the statistical analysis of these variables was only carried out with a subset of the data (Table 2).

The shedding of strongyle eggs across age groups showed significant differences (F = 16.55, df = 2, *p* < 0.001). Horses < 5 years old shed significantly more eggs (mean= 331.87 EPG, S.E. = 17, *p* < 0.01) (Table 3).

The current study revealed that the difference between the mean strongyle egg excretions for different genders was not significant. The prevalence of strongyle eggs in females (57.90%, mean = 258 EPG) was higher than in males (42.10%, mean = 250 EPG), but this difference was not statistically significant. However, the mean strongyle EPG was significantly higher in both females < 5 years (F = 14.98, df = 2, *p* < 0.001) and males < 5 years (F = 3.25, df = 2, *p* < 0.039) compared to older females and males (Table 3).

The mean strongyle EPG was higher for younger females (mean = 350.30 EPG, S.E. = 24.94) and males < 5 years old (mean = 323.79 EPG, S.E. = 27.43) than for older females and males, and these differences were statistically significant (*p* < 0.01) (Figure 2).

The season had a significant effect on strongyle egg shedding (F = 2.92, df = 3, *p* < 0.033) (Figure 3, Table 3), and approximately half of the animals were shedding strongyle eggs in different seasons (spring: 57%, both winter and autumn: 52%, summer: 48%).

The highest mean strongyle egg count was recorded in winter (305 EPG, S.E. = 29.27) when a high percentage of animals (33.8%) shed > 500 strongyle EPG. The lowest mean egg counts occurred in spring (146 EPG, S.E. = 46.80). The results highlighted that the differences in the mean strongyles EPG for winter and spring were statistically significant (*p* < 0.024) (Figure 3, Table 3). The comparison between regions did not reveal any significant differences.

A significant effect for the collection season in the different regions was observed (F = 4.06, df = 2, *p* < 0.017). In autumn, the difference in the strongyle EPG values for the southern region (357.62 EPG, S.E. = 41.96) and the north and western region (95.73 EPG, S.E. = 92.73) was statistically significant (*p* < 0.031). The highest mean strongyle EPG was found in the southern region (357.62 EPG, S.E. = 41.96), which was also statistically significant (*p* < 0.037), while the lowest mean EPG was found in spring in the same region (90.90 EPG, S.E. = 87.71) (Figure 4, Table 3).

## 4. Discussion

The risk of infection with GIH parasites is critical for the health of horses and their performance. Small strongyles have become the most important gastrointestinal parasite in horses over the past four decades [2,3]. This is the first study of Irish horses in which an attempt has been made to define the epidemiological aspects and risk factors that influence helminth infection and related egg shedding. Although the proportion of horses infected with GIHs is high across the world, ranging between 56 and 98%, it is generally rare to find animals with overt clinical disease [25,35,36,37,44,45,46]. The main helminth parasites observed in our study were strongyles, and the prevalence of infection ranged between 5 and 100% in horses reared with differing management practices. High prevalences of equine strongyle infection have been recorded in Brazil, Spain, Sweden, Italy, Lahore, the UK, Germany, Ethiopia, and the USA [25,35,36,37,44,45,46]. These findings are similar to the observation in the present study, where the prevalence of strongyle infection was 52.4% in horses on Irish farms. Overall, strongyle FECs from all 159 farms (n = 2700 animals) were relatively high, with a mean FEC of 250.22 EPG (range: 0–13,282.50 EPG). The mean strongyle FEC of 250.22 was higher than that reported in the UK (97 EPG) [24,35] but lower than that in Brazil and Ukraine (>500 EPG) [9,45]. This may be primarily related to the influence of the frequency of anthelmintic treatment on different farms, climatic factors, and other management practices. Most farms in this study treated their horses more than four times a year according to the data published by Elghryani, et al. [47].

Only 16% of animals shed > 500 EPG, which is similar to the findings of previous studies in the USA, where only 20% and 3% of Thoroughbred and Standardbred mares, respectively, shed > 500 strongyle EPG [25]; in the UK, 11% of horses shed strongyle eggs at the level of >500 EPG [35].

In this study, the prevalence of *Parascaris* spp. was low (4.22%), which is in agreement with studies from other countries [35,46,48,49]. The low prevalence of other helminth species, such as *Anoplocephala* spp. (2.59%) and *S. westeri* (0.9%), is also similar to other studies [16,48,49].

The FECs do not necessarily equate with low infection rates because FECs in general do not correlate well with the actual helminth burden [50,51]. It is possible to obtain an egg count of zero even when a horse is infected. This is based on necropsy findings, which showed that six horses carried large numbers of encysted cyathostomin larvae but had negative FECs. This can also be influenced by the minimum detection limit of the diagnostic techniques used [24,35,51,52,53]. In the present study, the more sensitive Mini-FLOTAC was used to estimate FECs. FECs are important to indicate the level of contamination of pastures and have been found to be useful as qualitative tools [40,54,55]. FECs can be influenced by many different factors, such as routine treatment with effective drugs and pasture and grazing management strategies. This is supported by studies that have investigated the effects of management practices on parasite burden [1,34,35,45,56].

Several studies have investigated the influence of the type of farm, animal stocking density, age, gender, anthelmintic used, and time since the last treatment on FECs. In our study, strongyle FECs were recorded in all age groups. However, the difference between the mean strongyle FECs was significantly (*p* < 0.001) greater in horses <5 years old compared to old animals (5–17 and >17 years). The results obtained confirm the findings of other studies in the UK [33,35], Sweden [57], Denmark [34], Ukraine [9], Spain [48], and Hungary [20], in which high FECs were found in animals <5 years of age. This is most likely to be related to the underdevelopment of the immune system at this age [58]. Horses <5 years old should receive particular attention when constructing parasite control programs. In contrast, another study found that old animals had higher FECs than young and mature animals [46]. This may have been due to deficiencies in the aged immune systems of these old horses.

This study found no significant difference in strongyle egg shedding in relation to gender; however, the results showed that the level of strongyle egg prevalence was high in females compared to males. This difference may reflect differences in grazing behaviour and management conditions, as shown in several other studies [35,46,48]. In addition, immunity status may also be related to sex hormones [59,60].

In the northern temperate climate, spring and summer provide favourable climatic conditions for the development and spread of infective-stage helminths on pasture. In the current study, strongyle infection was present in all seasons; however, there were differences in the levels of strongyle egg shedding in horses related to the seasons. The level of strongyle egg shedding was significantly (*p* < 0.001) lower in spring but increased during late winter, summer, and autumn, which is similar to findings in the UK [35]. The increase in strongyle egg shedding in late winter is most likely related to the emergence of encysted larvae during late autumn and early winter [61]. The reason for decreased strongyle egg shedding in spring may be related to anthelmintic treatment with moxidectin in late winter (December and January), as 83% of samples collected in spring were from horses that had been treated with moxidectin in late winter (unpublished data). This is in agreement with a previous study in the USA [21] that reported that horses treated with ML had lower FECs than those treated with other anthelminthics classes.

The prevalence of strongyle infection and the mean EPG across the country were similar. In the southern region, strongyle egg shedding dropped to its lowest level in the spring and started to increase during the summer, reaching its highest level in the autumn. The rising numbers of strongyle eggs are related to the greater contamination of pasture during summer (89% of horses graze > 16 h per day compared to 65% in autumn), as previously reported in Ireland [47].

## 5. Conclusions

Generally, strongyle eggs are ubiquitous in Irish horse farms and most common in horses <5 years old, and there were high mean EPG found for both genders. High prevalence was found in all seasons and regions; however, in winter, horses were significantly higher egg shedders. Based on this study, good management should be focused on and special awareness given to young horses, which are the higher risk group. Deworming programs with efficacious anthelminthics should be strategically tailored to particular farms and animal groups to adequately control strongyle infections in horses on Irish horse farms.

## Figures and Tables

**Figure 1 animals-13-00581-f001:**
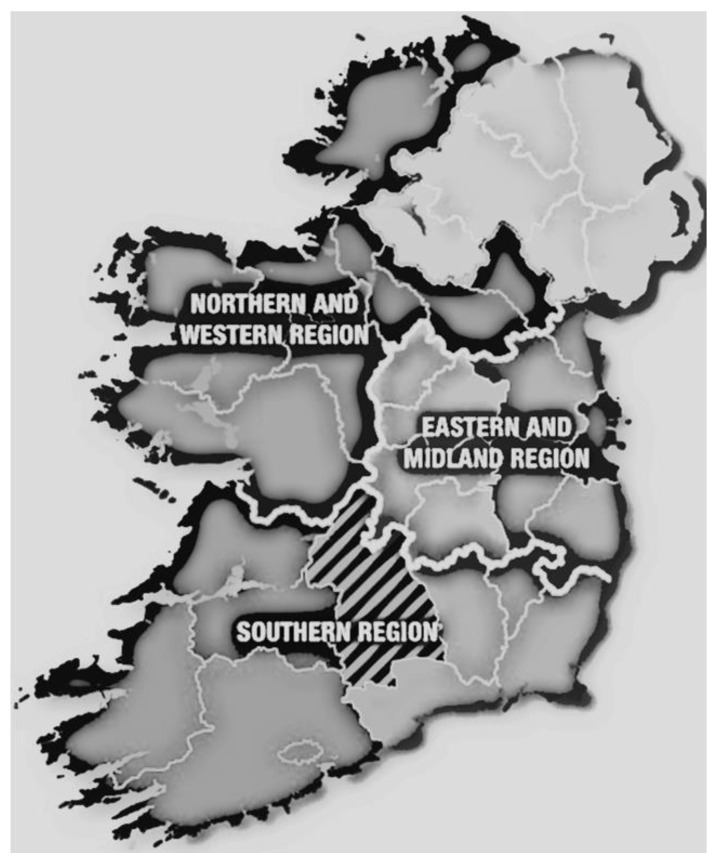
Map of regions in Ireland that was used in this study (https://www.nwra.ie/about/ (accessed on 28 November 2022)).

**Figure 2 animals-13-00581-f002:**
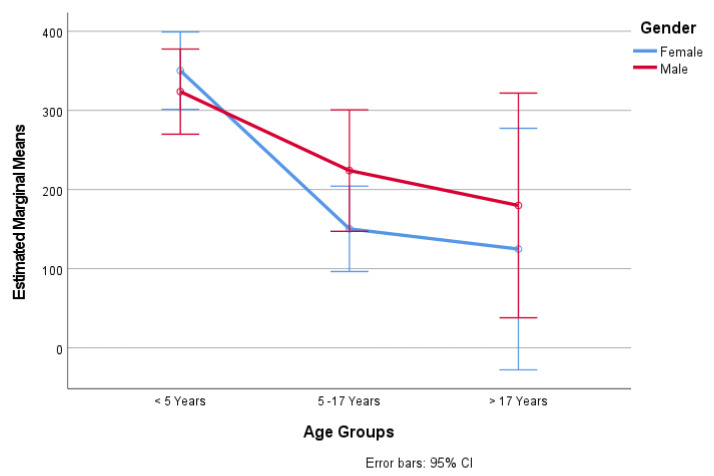
The estimated marginal mean strongyle egg counts for different genders and age groups.

**Figure 3 animals-13-00581-f003:**
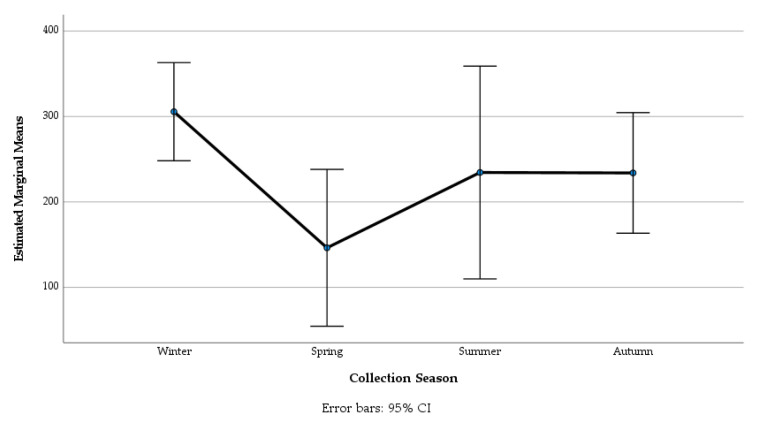
Estimated marginal mean strongyle egg counts during different seasons in horses in Ireland.

**Figure 4 animals-13-00581-f004:**
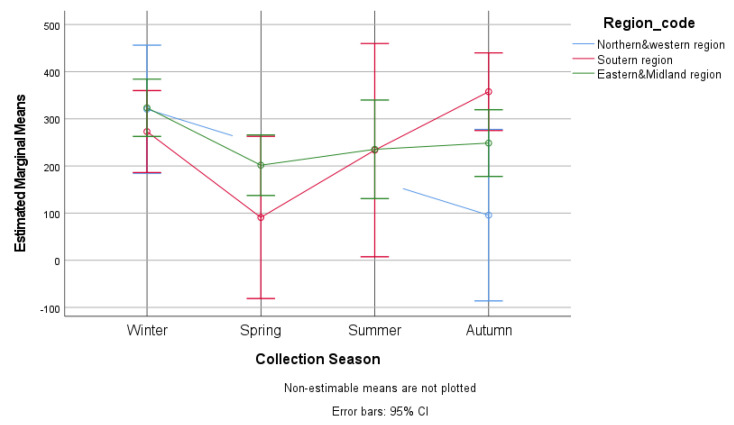
Estimated marginal mean strongyle eggs per gram for different collection seasons and geographical regions in Ireland.

**Table 1 animals-13-00581-t001:** Prevalence of infection with different helminth parasites in 2700 horses from Irish farms extrapolated from faecal egg counts.

Horses Infected with Various Helminth Parasites				
	Strongyle	*Parascaris* spp.	*Anoplocephala* spp.	*Strongyloides westeri*
Number of infected horses	1415	114	70	24
Prevalence %	52.40	4.22	2.59	0.89
Mean intensity (EPG) *	477.46	460.96	1	1
Std. error of mean	20.43	138.17	0	0
95% confidence interval	437.66–520.89	259–782	1	1

* Eggs per gram.

**Table 2 animals-13-00581-t002:** The prevalence percentages and descriptive statistics for strongyle eggs per gram (EPG) found in the examined horses reported for different factors.

Factors	Examined	Prevalence (%)	EPG MI ^1^ (S.E.) ^2^	95% CI ^3^
EPG level				
5–200	690	48.80	71.88 (2.17)	(67.42–76.46)
>200–500	295	20.80	329.04 (4.76)	(319.37–339.10)
>500 to 1000	228	16.10	712.37 (9.35)	(693.39–729.33)
>1000	202	14.30	1814.45 (84)	(1668–2001)
Total	1415			
Age group				
<5 years	904	39.91	331.88 (19.61)	(294.29–370.65)
5–17 years	544	15.72	180.55 (19.60)	(144.45–221.57)
>17 years	98	3.49	153.12 (22.07)	(96.18–219.95)
Total	1546			
Gender				
Female	1149	30.82	257.77 (15.25)	(229.12–286.36)
Male	908	22.41	250.33 (22.02)	(208.99–297.46)
Total	2057			
Area				
North and western region	95	64.21	240.06 (48.25)	(156.88–348.10)
Southern region	375	59.73	289.72 (35.74)	(226.08–366.54)
Eastern and midland region	928	58.19	258.13 (15.00)	(229.94–289.98)
Total	1398			
Season				
Winter (November–January)	711	52.90	305.68 (29)	(248.25–263.11)
Spring (February–April)	401	57.40	146.32 (46)	(54.99–238.14)
Summer (May–July)	376	47.00	234.52 (63)	(109.96–359.09)
Autumn (August–October)	1164	52.00	233.99 (35)	(163.39–304.60)
Total	2652			

^1^ MI: mean intensity, ^2^ S.E.: standard error of the mean, ^3^ CI: confidence interval.

**Table 3 animals-13-00581-t003:** Comparison of factors affecting the strongyle egg counts using a univariate general linear model. The factors that significantly affected the mean strongyle EPG (age and season) determined using a univariate general linear model.

Affected Variable	Mean Difference	Std. Error	95% Confidence Interval for the Difference	*p*-Value
Effect of age < 5 years old				
<5 by 5–17	194.83	30.25	77.33–222.35	<0.001
<5 by >17	184.65	56.27	49.76–329.53	0.003
Effect of the age inside the gender groups				
Within female group				
<5 by 5–17	199.94	37.13	110.94–288.93	<0.001
<5 by >17	225.46	81.7	29.62–421.29	0.018
Within male group				
<5 by 5–17	99.73	46.06	2.57–185.45	0.032
<5 by >17	143.83	62.22	8.58–257.96	0.027
Effect of collection season				
Winter by spring	159.36	55.21	13.44–305.23	0.024
Effect of collection season within each region				
Autumn: southern by north and western region	261.88	101.78	17.92–505.85	0.03
Autumn by spring in southern region	266.70	97.23	9.79–523.60	0.037

## Data Availability

The data presented in this study are available on request from the corresponding author. The data is not publicly available.

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
