# Peer review of "Estimating the Prevalence and Factors Affecting the Shedding of Helminth Eggs in Irish Equine Populations"

_animals, 2023, doi:10.3390/ani13040581_

Round 1

Reviewer 1 Report

This MS provides high quality data on helminth parasitism in horses in Ireland. The large sample size is commendable. I have a number of suggestions for improvement listed below.

Abstract:

Rather than GI parasites - specify helminths or even nematodes.

Natural distribution patterns - clarify.

Under key words include faecal egg counts (FEC).

Introduction:

Add a reference to the aggregated distribution of helminths (add to refs 29,30). Also see suggestion under results.

Materials and Methods:

Mention the number of horses in addition to the number of farms.

Was the faecal sample weighed and mixed prior to processing - if so, spell out.

Explain what estimates marginal mean is - is there a reference to this method? Which data were non normal?

Results: 

Explain the difference between mean epg and mean intensity (no zeroes?). Add to the methods.

Explain the 80/20 rule in Methods. I think the inclusion of the variance to mean ratio as a measure of aggregation would be a useful addition to the paper (see

Table 1: Explain the significance of the odds ratios in this Table - what is it telling the reader?

Table 2: It is not quite clear to me what is being compared in the epg factor analysis. Again you have odds ratios in the Table and F ratios in the text - explain. 

Table 3: This seems a rather cumbersome way of explaining these results. What about using interaction terms from the GLM?

I suggest omitting Figure 4. 

Author Response

The response for Review report 1

Abstract:

Rather than GI parasites - specify helminths or even nematodes.

  • Corrected to Helminths.

Natural distribution patterns - clarify.

  • Corrected to: helminth distribution patterns

Under keywords include faecal egg counts (FEC).

  • No need to define an abbreviation as this term is not used again in the abstract

Introduction:

Add a reference to the aggregated distribution of helminths (add

to refs 29,30). Also see suggestion under results.

  • Reference was added (5) (Nielsen, M.K.; Reinemeyer, C.R. Handbook of equine parasite control, Second ed.; John Wiley & Sons: 2018.)

Materials and Methods:

Mention the number of horses in addition to the number of farms.

  • Required information was added (a total of 2700 horses were randomly selected from 159 horse farms)

Was the faecal sample weighed and mixed prior to processing -

if so, spell out.

  • We added a brief explanation of the procedures

Explain what estimates marginal mean is - is there a reference

to this method? Which data were non normal?

  • The mean was estimated according to the estimates marginal mean, which gives estimates of predicted mean values in the model, and interaction plots of these means allow easy visualization of the relationships.

-      Reference:

(21) Nielsen, et.al. (2018 ) Risk factors associated with strongylid egg count prevalence and abundance in the United States equine population.

(40) Muller, K.E.; Stewart, P.W. (2006) Linear model theory: univariate, multivariate, and mixed models.

None of the data sets were normally distributed

  • Results:

Explain the difference between mean epg and mean intensity

(no zeroes?). Add to the methods.

Mean intensity (MI) was calculated as the mean number of EPG fund in the infected horses in a particular population), Std error and 95% Confidence Interval of the MI.

Explain the 80/20 rule in Methods. I think the inclusion of the

variance to mean ratio as a measure of aggregation would be auseful addition to the paper (see

For the 80/20 rule was explained in the methods.

For the variance-to-mean ratio, in our case, we expressed it as a standard deviation rather than a variance because the former is often more easily interpreted. also, the variance gives added weight to outliers. These are the numbers far from the mean. Squaring these numbers can skew the data.Table 1: Explain the significance of the odds ratios in this Table -

what is it telling the reader?

  • The odds ratios were deleted from the table but kept the significance of the results of the GLM, to make it easier for the reader to understand the outcome of the data.

Table 2: It is not quite clear to me what is being compared in the EPG factor analysis. Again you have odds ratios in the Table and F ratios in the text - explain.

  • In our results we compared the distribution of the strongyle EPG in relation to the factors examined.
  • When the F test showed significance, the GLM performed pairwise comparisons of the estimated marginal means of the dependent variables. These comparisons are performed among levels of a specified between- or within-subjects factor, and also performed separately within each level combination of other specified between- or within-subjects factors.

Table 3: This seems a rather cumbersome way of explaining

these results. What about using interaction terms from the GLM?

  • Table 3 is required to show significant data and easily visualize the relationship in the profile plots.

I suggest omitting Figure 4.

  • Deleted

Reviewer 2 Report

Detailed comments are given in the attached file

Author Response

The response for Review report 2

Comments on page 1:

strongyles, helminths or parasites? - Use one specific word through out the text according to your objective

  • Done: Helminths were used.

Should be faecal egg counts (FEC)

add full stop (spp.) correct it through out the text

  • Spp. Was corrected through out the text.

Should be egg per gram (EPG)

  • Egg per gram was added

Only strongyle? - Use the correct word

  • Line 30: the analysis was done for strongyles as the data of other all the other helminths were very low.

Must be Parascaris (P.) equorum, P. univalens, Strongyloides (S.) westeri, Anoplocephala spp and Oxyuris (O.) equi

  • Line 38: In general scientific writing it is not necessary to define the abbreviation of the genus names. However, if required by the editor we will make this change.

add specific references rather than a long list

 Done. Reference was added (5) (Nielsen, M.K.; Reinemeyer, C.R. Handbook of equine parasite control, Second ed.; John Wiley & Sons: 2018)

Comments on page 2:

add very relevant and specific references

- Done. Reference was added (18) (Nielsen, M.K. Anthelmintic resistance in equine nematodes: Current status and emerging trends. Int J Parasitol Drugs Drug Resist 2022, 20, 76-88, doi:10.1016/j.ijpddr.2022.10.005)

Sentences should be short, meaningful, and comprehensive

  • The text have been updated to make it more precise

Only cyathostomins?

  • Corrected to helminths.

Comments on page 3:

what was the sampling criteria? Mention clearly

  • More information on the sampling criteria was added.

delete space

  • Not done. In scientific writing a space is inserted between number and the unit.

Comments on page 4:

It is better to describe the technique briefly

  • More information was added.

Reference format should be according to the Journal guidelines

  •  

what was the criteria to identify the eggs of specific species?

  • These were different genera of different helminths that can easily be recognised by both the shape and the measurement of the egg.

It should be under the heading of sample collection

  • Added to sample collection.

How this test is suitable for your study? Justify with a suitable reference 

Why did you not use Chi square test for this study

  • Information and references were added.
  • the GLM performed pairwise comparisons of the estimated marginal means of the dependent variables. These comparisons are performed among levels of a specified between- or within-subjects factor, and also performed separately within each level combination of other specified between- or within-subjects factors, which can not be done by Chi square.

-      Reference:

(21) Nielsen, et.al. (2018 ) Risk factors associated with strongylid egg count prevalence and abundance in the United States equine population.

(40) Muller, K.E.; Stewart, P.W. (2006) Linear model theory: univariate, multivariate, and mixed models.

  •  

mean of which ?

  • Mean of strongyle EPG

Comments on page 5:

The number of examined animals are the not same under each variable, Why?

  • This was mentioned in the relevant text above of Table 2. – because of missing information for some animals.

Why did you not compare the results of other prevalent parasites

  • This was mentioned in the relevant text (line 140,141), the analysis was done with strongyles as the prevalence of other helminths were very low

Comments on page 9:

should be part of results

  • Already mentioned in the results.

should be part of results

  • Moved to results.

 Comments on page 10:

What is the relation of this paragraph with your objective?

did you compare the results of McMaster and Mini-FLOTAC technique?

  • the paragraph was adjusted

 we are investigated the prevalence of helminths EPG which can be influenced by many different factors starting from the technique we used,

Round 2

Reviewer 2 Report

In Table 2 add following columns? Number of positive, p value.

Sample size is too much confusing.

Total sample collection = 2700

Samples from different regions = 1398

Samples of different age group = 1546

Samples of different genders = 1639

Samples of different seasons = 2652

Why sample size of each variable is different? Sample size of each variable should be equal to total sample collection.

After correction of numbers of sample of each variable, results should be updated accordingly 

Author Response

The response for Review report 2

Comment 1:

In Table 2 add following columns? Number of positive, p value.

- Table 2 is showing only the prevalence percentages and descriptive statistics of strongyle eggs per gram EPG found in examined horses reported for different factors.  To further remove any ambiguity we also add the text to a footnote of Table 2. 

- The positive P-value was presented in table 3.

Comment 2

Sample size is too much confusing.

Total sample collection = 2700

Samples from different regions = 1398

Samples of different age group = 1546

Samples of different genders = 1639

Samples of different seasons = 2652

Why sample size of each variable is different? Sample size of each variable should be equal to total sample collection.

After correction of numbers of sample of each variable, results should be updated accordingly 

- We mention in the methods that “Due to missing information for some animals, the statistical analysis of these variables was only carried out on a subset of the data. Table 2”.

- The general linear model univariate (GLM) test can handle empty cell problems when we run a pairwise comparison of the estimated marginal means of the dependent variables (strongyle EPG) to evaluate the differences among specific means of EPG between factors, the GLM excludes any incomplete case.

Round 3

Reviewer 2 Report

The authors are unable to justify the statistical analyses and sample size variation of different variables. 

Author Response

The response for the review report 2

Comments and Suggestions for Authors:

The authors are unable to justify the statistical analyses and sample size variation of different variables. 

  • The complete case analysis is defined as an analysis restricted to individuals with complete information on all variables used in the analysis. This approach can result in biased results so it is never the best way. It is now common practice to use general liner models that are better able to handle analysis with some missing data points. This was then also our approach in this paper. Table 2 was also updated to show the number of samples analysed. Below is some paper that use different variable analysis
  • It’s a common approach as the references here:
  1. Joó, K.; Trúzsi, R.L.; Kálmán, C.Z.; Ács, V.; Jakab, S.; Bába, A.; Nielsen, M.K. Evaluation of risk factors affecting strongylid egg shedding on Hungarian horse farms. Veterinary Parasitology: Regional Studies and Reports 2022, 27, 100663, doi:https://doi.org/10.1016/j.vprsr.2021.100663.
  2. Scala, A.; Tamponi, C.; Sanna, G.; Predieri, G.; Dessì, G.; Sedda, G.; Buono, F.; Cappai, M.G.; Veneziano, V.; Varcasia, A. Gastrointestinal Strongyles Egg Excretion in Relation to Age, Gender, and Management of Horses in Italy. Animals 2020, 10, 2283.
  3. Belay, W.; Teshome, D.; Abiye, A. Study on the Prevalance of Gastrointestinal Helminthes Infection in Equines in and around Kombolcha. Journal of Veterinary Science and Technology 2016, 2016.
  4. Matto, T.N.; Bharkad, G.P.; Bhat, S.A. Prevalence of gastrointestinal helminth parasites of equids from organized farms of Mumbai and Pune. Journal of Parasitic Diseases 2015, 39, 179-185, doi:10.1007/s12639-013-0315-4.
  5. Schneider, S.; Pfister, K.; Becher, A.M.; Scheuerle, M.C. Strongyle infections and parasitic control strategies in German horses - a risk assessment. BMC Vet Res 2014, 10, 262, doi:10.1186/s12917-014-0262-z.
  6. Saeed, K.; Qadir, Z.; Ashraf, K.; Ahmad, N. Role of intrinsic and extrinsic epidemiological factors on strongylosis in horses. Journal of Animal and Plant Sciences 2010, 20, 277-280.
  7. Lloyd, S. Effects of previous control programmes on the proportion of horses shedding small numbers of strongyle-type eggs. Vet Rec 2009, 164, 108-111, doi:10.1136/vr.164.4.108.